# The Significance of the Response: Beyond the Mechanics of DNA Damage and Repair—Physiological, Genetic, and Systemic Aspects of Radiosensitivity in Higher Organisms

**DOI:** 10.3390/ijms26010257

**Published:** 2024-12-30

**Authors:** Peter V. Ostoich

**Affiliations:** Institute of Biodiversity and Ecosystem Research, Bulgarian Academy of Sciences, 1113 Sofia, Bulgaria; p.ostoich@gmail.com

**Keywords:** radiobiology, DNA damage and repair, mutation, radiosensitivity, comparative genetics, genetic disorders, signaling, immunomodulation

## Abstract

Classical radiation biology as we understand it clearly identifies genomic DNA as the primary target of ionizing radiation. The evidence appears rock-solid: ionizing radiation typically induces DSBs with a yield of ~30 per cell per Gy, and unrepaired DSBs are a very cytotoxic lesion. We know very well the kinetics of induction and repair of different types of DNA damage in different organisms and cell lines. And yet, higher organisms differ in their radiation sensitivity—humans can be unpredictably radiosensitive during radiotherapy; this can be due to genetic defects (e.g., ataxia telangiectasia (AT), Fanconi anemia, Nijmegen breakage syndrome (NBS), and the xeroderma pigmentosum spectrum, among others) but most often is unexplained. Among other mammals, goats (*Capra hircus*) appear to be very radiosensitive (LD_50_ = 2.4 Gy), while Mongolian gerbils (*Meriones unguiculatus*) are radioresistant and withstand quadruple that dose (LD_50_ = 10 Gy). Primary radiation lethality in mammals is due most often to hematopoietic insufficiency, which is, in the words of Dr. Theodor Fliedner, one of the pioneers of radiation hematology, “a disturbance in cellular kinetics”. And yet, what makes one cell type, or one particular organism, more sensitive to ionizing radiation? The origins of radiosensitivity go above and beyond the empirical evidence and models of DNA damage and repair—as scientists, we must consider other phenomena: the radiation-induced bystander effect (RIBE), abscopal effects, and, of course, genomic instability and immunomodulation. It seems that radiosensitivity is not entirely determined by the mathematics of DNA damage and repair, and it is conceivable that radiation biology may benefit from an informed enquiry into physiology and organism-level signaling affecting radiation responses. The current article is a review of several key aspects of radiosensitivity beyond DNA damage induction and repair; it presents evidence supporting new potential venues of research for radiation biologists.

## 1. Introduction

Ionizing radiation (IR) is probably the best-studied genotoxic agent in our environment, with vast amounts of data underpinning our current understanding of the damage it induces and its repair. Remarkable for its low energy-to-effect ratio (an LD_50_ dose, delivered to an average 70 kg person, is just 4 Gray (Gy, 1 Gy = 1 J/kg), or, in terms of calories, 67 cal [1]. Thermodynamically negligible (and undetectable by our senses) amounts of energy in the form of IR induce DNA damage, which alters the cellular kinetics, and can kill, at high doses, predictably (or “deterministically”), by inducing acute radiation syndrome and, at lower doses, “stochastically”, by causing malignancies. This first section discusses classical radiation biology and its relationship with the mathematics of DNA damage and repair, both as a foundation for our current understanding of radiation damage and as an explanation to why we, as scientists, may not always be asking the right questions.

Before even the Avery experiment demonstrated in 1944 that genomic DNA was (back then, this seemed surprising) the main carrier of hereditary information in organisms [2], scientists and medical doctors knew for certain that different organs and cell types responded in different ways to exposure to ionizing radiation (IR). In 1906, French X-ray pioneers observed “higher effects of Röntgen-ray exposure on fast-dividing cells”; this became known as the law of Bergonié and Tribondeau [3,4]. In 1911, during the dawn of radiation therapy for skin diseases and cancer, Claudius Regaud had demonstrated—by sterilizing rams without causing skin burns—that fast-dividing cells subjected to protracted discrete-dose exposure are more sensitive to X-rays [5]. This became the basis for present-day fractionated radiotherapy for cancer, based on the linear-quadratic model stipulating increasing differences in survival between normal-tissue (NT) and tumor cell types during fractionated exposure [6,7]. Later, in the post-Manhattan Project era, and with the advent of early cell culturing, cell damage and the kinetics of cell division were directly related to radiation exposure, with some phases of the cell cycle (late G1, G2, and the M phase) being more sensitive to equivalent energies and deposition patterns [8,9]. The development of sufficiently precise mammalian cell cultures introduced a completely new way of thinking about radiation damage, namely as manifestations of killing of target cells characterized by specific survival curves, typically plotted on a logarithmic scale [8]. These pre-digital investigations and models of radiation damage and repair can best be summarized in the words of M. M. Elkind: “Three dualities underscore the role of repair in radiobiology”: the *discrete* and *random* nature of energy deposition (“all molecules in a cell are susceptible to radiation damage, but only a few are hit”); the coupled processes of damage and repair; and the harmful and beneficial properties of radiation in connection with issues of public health and the radiation therapy of cancer, respectively [10].

For several decades now, classical radiation biology has walked hand in hand with the mathematics of DNA damage and repair. Although “modern radiation biology” after the year 2000 has accounted for some of the phenomena described below, such as radiation-induced bystander effect (RIBE), abscopal effects, and radiation-induced genomic instability, scientific risk estimates are still based on the “linear non-threshold” (LNT) derived from the dose-dependent effects and mathematics of classical radiation biology [1]. When we think about radiation risk, genomic DNA is still the primary target. Abundant experimental evidence confirms that radiation-sensitive areas of the cell are located in the nucleus as opposed to the cytoplasm; classical experiments with α-irradiation (a densely ionizing radiation modality, with each particle delivering >1 Gy to a cell nucleus) demonstrated that cytoplasmic irradiation with colossal doses (>200 Gy) has no influence on cell proliferation [11]. Radiation induces a large number of lesions in DNA, and the vast majority of them are repaired successfully by the cell. A dose of radiation that induces an average of one lethal event per cell (mean lethal dose or D_0_ dose) leaves 37% of a cell population still viable [1]. The D_0_ dose for mammalian cells usually lies between 1 and 2 Gy. Depending on radiation modalities, the number of DNA lesions per cell detected immediately after such a dose is approximately the following:40 double-strand DNA breaks (DSBs);~1000 single-strand DNA breaks (SSBs);>1000 events of DNA base damage [11].

Despite being the least numerous of these lesion types, DNA DSBs correlate best with cell killing and require the longest time to resolve; reactive oxygen species such as H_2_O_2_ (or, for that matter, mobile phone radiofrequencies) induce vast amounts of SSBs with little impact on mammalian cell survival [12,13]. As a background, each mammalian cell experiences ~70,000 DNA damage events per cell every day, with thousands being SSBs, and these lesion types have a very short biological half-life due to effective SSB repair [14].

Conversely, DSBs are the most cytotoxic type of primary DNA lesion. Unrepaired or defectively repaired DSBs cause mutations or loss of chromosome regions, eventually leading to cell death or neoplastic transformation, with cell death being caused by (1) induction of chromosomal aberrations; (2) lethal mutations; and (3) induction of apoptosis [12,13,14,15,16,17]. It is widely accepted that one unrepaired DNA DSB is usually lethal to a mammalian cell [18]. Perhaps fittingly, evolution has supplied higher organisms with seven (or eight, depending on the source) main DNA repair systems, of which three are dedicated to DSB repair:**Homologous recombination (HR)**, involving notably ATM, BRCA1/2, Rad51, Rad17/RFC, NBS1, etc.**Classical non-homologous end-joining (NHEJ)**, dependent on DNA-PK, Ku70/Ku80, Artemis, and XRCC4/Ligase 4, as well as ATM.**Microhomology-mediated end joining (MMEJ), also known as alternative non-homologous end-joining (Alt-NHEJ)**, involving PARP-1, NBS1, Rad50, MRE11, and XRCC1 and recruiting PolQ and Ligases 1 and 3.

In addition to these three DNA repair systems, dedicated to DSB repair, there are at least four other distinct repair systems, dealing with specific less-significant types of damage:**Base excision repair (BER)**, dealing with single-strand breaks (SSBs) and DNA base oxidation and involving PARP-1, XRCC1, Polβ, and Ligase 3;**Nucleotide excision repair (NER)**, which resolves DNA adducts and pyrimidine dimers resulting from ultraviolet (UV) exposure; involves XPA, XPB, XPC, PCNA, and ERCC1/XPF; and recruits Polymerases δ (delta) and ε (epsilon) and Ligases 1 and 3;**Mismatch repair (MMR)**, repairing replication errors and random point mutations and involving MSH2, MSH6, MLH1, the Exo1 (HEX-1) exonuclease, and Polymerase δ/Ligase 1;**Interstrand cross-link repair (ICLR)**, utilizing FANCM, FANCD, and the multimeric Fanconi Anemia protein complex (comprising FANCA, FANCB, FANCC, FANCF, FANCG, etc.), and Rad51C/SLX4 as a downstream gateway leading to either NER or DSB repair [1,12,14,15].

Nevertheless, the most significant (and most deleterious) type of DNA damage in mammalian cells remain double-strand breaks (DSBs).

When it comes to DSB induction and repair, not all radiation modalities are “created equal”. Energy deposition densities, mechanisms of interaction with matter, and, therefore, inducible DNA damage, tissue penetration, and overall biological effects depend on the type, energy, and dose rate of the incident ray. “Sparsely ionizing” radiations, such as photons (γ- and X-rays) and electrons (β-particles), deposit their energy randomly and, to a large extent, via creation of reactive oxygen species (ROS); they have an “oxygen enhancement ratio” (OER) exceeding 2.5, implying that ~70% of the induced biomolecular damage is indirect and due to free oxygen radicals [1]. In contrast, “densely ionizing” radiations, such as α-particles and some types of cosmic radiation (e.g., heavy ions), deposit most of their energy directly (OER = 1), leading, in practice, to very high localized energy depositions and clustered DNA damage; on a cellular level, there are no “low doses” for α-particles, since traversing the nucleus by a single α-particle nearly always produces an equivalent dose exceeding 1 Sievert (Sv) [19,20]. Depending on their energy and type, different radiation modalities have different deposition patterns in tissue, characterized in part by “linear energy transfer” (LET, expressed as loss of energy over distance, keV/µm). For example, charged particles, such as protons (*p*+) and heavy ions (e.g., cosmic or accelerator-derived helium nuclei, carbon ions, and neon ions), tend to deposit most of their energy at a certain depth in tissue in a phenomenon called a Bragg peak [1,20]. Although photonic IR modalities, such as γ- and X-rays, deposit their energy in a similar way via photoelectric effect (PE), Compton scattering, and, at energies above 1.022 MeV, pair production (formation of one electron and one positron), their biological effects can vary depending on the energy spectrum of the photons [1]. Notably, softer X-rays (40–50 kVp range) have lower penetration but higher LETs in comparison with ^60^Co (1.17 and 1.33 MeV γ-rays), leading scientists to designate them with a higher “relative biological effectiveness” (RBE) of 1.5–1.6 [21]; the pre-isotope source standard for high-power X-ray tubes (250 kVp) is now widely accepted to have an RBE of 1.1–1.2 in comparison with ^60^Co photons [22].

The mathematics of DNA damage and repair seem to work out in most cases. We know this from more precise immunofluorescent methods for the visualization of induced and residual DSBs per unit dose in the last two decades such as (1) γH2AX foci induction (focal phosphorylation of histone H2AX at residue 139, which occurs at DSB sites and is detectable by antibodies) [23] and (2) formation of discrete 53BP1 foci in the cell nucleus, which dissolve during repair and resolution of mammalian DSBs [24]. Rothkamm and Löbrich [25] postulated sensitivities of the γH2AX assay in the very low dose range (1–5 mGy) with standardized cell cultures and good microscopy, and, while this has been debated, it is certain that γH2AX counting can distinguish DSB induction at steps of 20–25 mGy with statistical significance, while 53BP1 staining and counting can yield good data on DSB repair kinetics. Perhaps most significantly, residual DNA damage leads to chromosomal aberrations in lymphocytes (mostly dicentrics but also ring chromosomes), which decrease over time, and can be used for biodosimetry (cytogenetic reverse dose-estimation of accidentally exposed individuals) with a high degree of precision, typically below 0.25 Sv [26,27]. If the mathematics work out both forward and in reverse, how can classical radiation biology be problematic?

## 2. Challenges to Classical Radiation Biology and the Current Risk Estimates Derived from It

### 2.1. All Species Are Created… Different

It is well known that some invertebrate animal species are fantastically radioresistant. Tardigrades can withstand 5000 Gray with 50% loss in viability (LD_50_ = 5000 Gy) [28]. For comparison, the LD_50_ for humans (without medical intervention) is around 4–4.5 Gy, and for mice it is around 6.4 Gy. Among other mammals, goats (*Capra hircus*) appear to be very radiosensitive (LD_50_ = 2.4 Gy), while Mongolian gerbils (*Meriones unguiculatus*) are radioresistant and withstand quadruple that dose (LD_50_ = 10 Gy) [29]. And, while at doses above 1 Gy, ARS proceeds similarly in all investigated mammalian species (with the exception of humans, who are the only organism that invariably vomits upon ARS-inducing exposure) [1,29], induction of neoplastic disease appears differently in each distinct species.

Experimental evidence becomes conflicting and curious observations start to occur when moving into the lower dose range. At whole-body doses below 0.7–1 Gy, which are insufficient to induce acute radiation syndrome (ARS), the main risk of radiation lethality is *stochastic*, meaning an increase in the probability of solid tumors and hematological malignancies. Each individual species develops different types of cancers upon radiation exposure. For example, most human solid tumors, including radiogenic ones, are carcinomas; they come from epithelial cell types (including skin, mucosa, intestinal epithelium, mammary duct epithelium, etc.) [30,31,32]. Unlike humans, mice (which are >95% genetically identical to humans) predominantly develop sarcomas (connective tissue tumors) [33,34]. Still more telling is the fact that tumor suppressor genes act differently in mice and men: for the Rb gene, heterozygous deficiency in humans causes retinoblastomas and osteosarcomas, while the same deficiency in mice has no observable effects on the phenotype [35,36,37,38]. In humans, strong expression of the cell cycle arrestor p15 (known alternatively as Cyclin-dependent kinase 4 inhibitor B, multiple tumor suppressor 2 (MTS-2), or p15^INK4b^) in connective tissue has been linked to (1) resistance of human connective tissue to carcinogenesis and (2) strict maintenance of the Hayflick limit in human fibroblasts in vitro [30,32,39]. Surprisingly, mice with a complete ablation of both p15(INK4b) alleles had a lower tumor incidence and higher survival rate when compared with those with homozygous or heterozygous expression of p15(INK4b) [40]. The abovementioned differences are just the tip of the iceberg, demonstrating that the radiation responses of organisms are essentially different and that the exact same gene (and gene product) often has completely different effects in two organisms as similar as mice and men.

### 2.2. Radiation Sometimes Causes Damage Very Far from Where It Hits

A well-established late sequela of any form of radiotherapy (RT) is the possibility of inducing a radiogenic primary cancer within, or near, the treatment field, with nearly 80% of the resulting cancers curiously arising from low-exposure areas receiving <6 Gy [41]. Similar risks accompany high and repeated diagnostic exposures and exposure to atomic weapon radiation (Hiroshima and Nagasaki Lifespan Study (LSS) cohorts) [42,43]. Where the science really becomes surprising is that radiotherapy for lung cancer significantly increases the risk for second primary bladder cancers and colorectal cancers (CRCs) by 62% and 74%, respectively, confirmed by large-cohort epidemiology based on the US SEER database [44]. In any case of competent RT, these organs are sufficiently outside the treatment field and receive negligible doses.

This “distal carcinogenesis” sometimes, admittedly rarely, works in reverse. “Abscopal effects” of RT refers to the shrinkage or complete elimination of distal metastases following localized irradiation of a primary tumor; with fewer than 50 confirmed cases, this is a comparatively rare phenomenon [45]. Nevertheless, a clinical explanation exists: the vast majority of abscopal cures were registered in patients with “immunologically hot” (i.e., targeted by the immune system) tumor types; the probable explanations are reactivation of the immune system (particularly Type I interferon signaling and cytotoxic T-cell activation) and more generalized “bystander signaling” (more on that phenomenon in the following sections) [45,46,47].

### 2.3. Non-DNA Targets and Non-Target Effects of Ionizing Radiation

Classical radiation biology identified genomic DNA as the main, if not only, target for significant radiation damage [1,11]. What if that was only part of the story? Researchers have looked at other biomolecules outside the cell nucleus, particularly mitochondrial DNA (mtDNA) and membrane lipids. Human mtDNA (16,564 base pairs) is the size of about 1/400 (0.25%) of human genomic DNA; there is less efficient repair in mtDNA than in genomic DNA [48]. The repair of mtDNA includes mainly BER components of MMR and alternative NHEJ repair pathways, with no evidence of NHEJ and questionable involvement of HR [49,50]. Escaping damaged DNA from mitochondria can promote inflammatory responses; “leaky mitochondria”, in general, contribute to intracellular ROS and lead to apoptosis [49,50]. Similarly, radiation damage can lead to easily detectable oxidation of cell membranes (formation of malondialdehyde and related products), which tends to correlate well with apoptosis [51]. Nevertheless, both mitochondrial instability and membrane damage are probably related to radiation sensitivity indirectly and via cell signaling such as inflammatory responses and death receptors like FAS, TNF, CD40, and TRAIL [50,52]. This means, in practice, that in the low dose range (relevant for radioprotection), mtDNA and cell membranes are secondary targets, only to be considered as signal initiators in DNA damage responses (DDRs) and apoptosis; in the high dose range, there will always be the “hen-and-egg problem”: leaky mitochondria and damaged cell membranes initiate pro-apoptotic signaling, but they are also a result of apoptotic cascades initiated by the very same signaling.

*(a)* 
*Cell signaling as the target: the radiation-induced bystander effect (RIBE)*


Anecdotal evidence from the period of 1960 to 1990 suggested that culture medium from irradiated cells could induce DNA damage; in a classical experiment of radiation biology, Nagasawa and Little [53] demonstrated in 1992 that very low doses of 3.7 MeV α-particles, leading to <1% of cell nuclei being hit, induced sister chromatid exchanges in ~15% of all chromosomes in CHO cells. This type of experiment has been re-iterated repeatedly in different settings (for instance, in vitro with partially shielded culture flasks or transfer of culture medium, or in vivo in mice and zebrafish), and this “radiation bystander effect” (RIBE) has been detected in a number of experimental systems, fascinating radiation biologists [54,55,56,57,58]. The RIBE phenomenon was one of the central topics of the European NOTE consortium and project (2006–2010), and although the implications of RIBE for low-dose risk are still uncertain, “the cat is out of the bag” and RIBE-associated cell signaling is a valid and pertinent research question [58,59].

Central to bystander signaling in mammalian cells in vitro appear to be three phenomena: (1) the bystander response becomes saturated at relatively low doses (typically less than 1 Gy) [46,60]; (2) bystander effects in vitro act via small molecules (such as nitric oxide, NO) or larger “messengers”: TNF-α, IL-1α, IL-1β, IL-6, IL-8, TGF-β, and TRAIL [46]; (3) all of these effects depend on serum supplementation, with the curious observation that the most critical serum component for the induction of bystander responses in mammals is serotonin [61,62]. Bystander effects in vivo probably work by interfering with cellular DNA repair and DDR, as well as by subjecting cells to faulty death signaling [56,57,58]. Despite the fact that bystander responses undoubtedly exist, they remain enigmatic, and their relevance for clinical medicine and radioprotection remains an unanswered question; still, they continue to provide a valid demonstration that there is one other confirmed target for ionizing radiation in addition to genomic DNA: cell signaling.

*(b)* 
*Gene maintenance as the target: radiation-induced genomic instability*


It is axiomatic that different organisms have divergent spontaneous mutation frequencies stemming from variety in replication fidelity and DNA repair competence; some polymerases like *Pfu*, derived from *Pyrococcus furiosus*, are high-fidelity, stable molecules; viral RNA-dependent RNA polymerases are usually low-fidelity enzymes, providing evolution-driving high mutation frequencies at the cost of virion viability [63]. Likewise, different individuals of the same species can have widely varying spontaneous mutation frequencies (discussed at greater length below under “Syndromes and defects inducing radiosensitivity”). Surprisingly, exposing parental organisms to significant doses of genotoxins has sometimes led to increased mutation frequency in the offspring, an “induced genomic instability”. Cadmium (Cd), a toxic metal, is well known to cause this in the progeny of exposed organisms, from *Arabidopsis thaliana* to mammalian cells in vitro to different mouse strains [64,65].

Evidence that radiation can induce transmissible genomic instability in the offspring of irradiated cells or organisms over many generations has steadily accumulated, starting from an observation by Kennedy et al. (1980) that X-ray exposure increases malignant transformation frequency in the distant progeny of irradiated C3H 10T1/2 cells [66]. Admittedly, this cell line is not a very good model for normal genome maintenance—C3H 10T1/2 are a fibroblast-like embryonic line stemming from C3H mice, which are prone to developing cancer, especially mammary gland tumors, as well as muscular dystrophy and autoimmune diseases [67]. In the mid-1990s, Yuri Dubrova and colleagues detected increased minisatellite instability in the children of Chernobyl-exposed populations [68,69]. These observations have been confirmed in mice [70], and although contested (usually with the argument that minisatellite DNA is non-coding and has a high spontaneous mutation frequency), remain a valid challenge for radioprotection; proposed mechanisms include “latent DNA damage” or “epigenetic switching” in the germline cells.

### 2.4. Conditions and Genes Affecting Sensitivity to Radiation

Cancer radiotherapists understand well the tradeoff between the doses needed to eradicate a tumor and the doses that induce prohibitive acute effects or cause life-changing late sequelae of the treatment. Clinical studies of radiotherapy side effects have demonstrated that a large part of the spectrum of normal-tissue reactions, perhaps as much as 80%, is due to differences in individual normal-tissue (NT) sensitivity; evidence suggests that the most “resistant” 40% of patients could be dose-escalated by 17–18%, which is likely to be associated with significant gains in control of the primary tumor, perhaps by as much as 34–36% [71]. On the other hand, early toxicities during therapy (occurring primarily in turnover tissues (e.g., bone marrow, epidermis, mucosae of the gastrointestinal tract)) prevent effective cures, and an estimated 18% of adult survivors cope with permanent consequences [72,73,74,75]. The individual normal-tissue radiosensitivity of patients typically follows a normal distribution, with most “IR sensitivity cases” of the values to the end of the peak being unexplained except the “extreme left” cases attributable to known genetic deficiencies [76,77,78]; likewise, there is no coherent theory on above-average radioresistance, although anecdotal evidence suggests rare but extreme limits in humans, such as hematopoietic (and general) recovery after total-body exposure > 10 Gy, which seems fantastic due to the near-universal lethality of intestinal damage above 8 Gy [79].

Still, it may be of merit to discuss known factors inducing normal-tissue radiation sensitivity in humans. Some of the best-known genetic defects leading to increased somatic effects of radiation are presented in Table 1.

As follows from the data presented, most of these conditions are recessive genetic diseases affecting DNA repair systems (often DSB repair), with syndromes inducing serious radiosensitivity (AT, NBS, etc.) affecting two or more repair systems. Three of these conditions are defects of helicases bearing homology to the bacterial RecQ (Werner, Bloom, and Rothmund–Thomson syndrome), with Werner syndrome presenting an interesting case: an increase in cancers that are typically very rare in patients (hemangiomas, meningiomas, and myelodysplastic syndrome (MDS)). Due to arguments discussed earlier (i.e., different types of cancers dominating statistically in different species) [30,31,32,33,34], the dynamics of WRN (the effector mutated) may hold the key to why these cancers appear rarely in humans. Likewise, progeria presents an interesting case: Lamin A is seldom thought of as key to DNA repair, being mostly considered structural for nuclei and chromosomes. Nevertheless, deficiencies in Lamin A, such as HGPS and restrictive dermopathy (RD), induce a premature senescence phenotype and susceptibility to DNA damage [84].

Unlike radiosensitivity syndromes, heritable predisposition to cancer (including radiogenic cancer) is not always directly related to the DNA repair machinery. Heritable predisposition to cancer is more likely to be autosomal dominant, and conditions like Li–Fraumeni syndrome and hereditary retinoblastoma (p53 and RB1 heterozygous defects, respectively) are due to a deficit in a single copy of a cell cycle “handbrake” acting in response to DNA damage. Li–Fraumeni patients are notoriously susceptible to radiogenic tumors despite bearing only a single defective copy of the *p53* gene; curiously, p53 is required for telomere-associated senescence and the related pro-oxidative state and DNA damage response [85,86]. This would, theoretically, make p53 both a tumor suppressor and tumor promoter [30,32,86,87]. Some of the best-known hereditary cancer predisposition syndromes are presented in Table 2.

General predisposition to tumors and susceptibility to radiogenic malignancies presents a diverse picture; in these and other conditions, the etiology of cancer seems to be much more closely related to a signaling component than to DNA repair fidelity and capacity. Still, even in the area of high-throughput sequencing, industrial genomics, and AI, scientists have only seen the surface of the genetic basis of radiosensitivity and radioresistance.

### 2.5. Radiation as an Immunomodulator: Yesterday’s Lessons Are Relevant Today

In the earliest days of radiotherapy, dating back to Claudius Regaud in Paris and Robert Kienböck in Vienna, cancer treatment by X-rays was experimental. Other treatments were not: X-irradiation for psoriasis, hairy nevus, scalp ringworm (Tinea capitis), abscesses and the occasional palliative irradiation of an arthritic knee were widely accepted and standardized medical practice [88,89,90,91,92]. Radiotherapy for benign diseases is still an option today [88], although doctors are generally reluctant to commit a therapeutic dose for a non-lethal disease; more than anything else, non-oncological radiotherapy has been tainted by the “Ringworm affairs” in Israel (1948–1960) and postwar Yugoslavia, (1950–1959, supported by UNICEF) leading to thousands of children being irradiated and suffering later in life [91,92]. In the pre-antibiotic era, before the late 1950s, radiation therapy saved lives from acute postpartum mastitis (APM) infections and commonly alleviated the disabling symptoms of ankylosing spondylitis (Bekhterev’s disease) at the cost of increased risk of cancer in later life [93,94]. The mechanisms for the anti-inflammatory effects of radiation are not entirely understood, but observations indicate increased expression of the anti-inflammatory cytokine interleukin 10 and downregulation of TNF-α in lymphocytes, with reduced expression of E-selectin and an increased induction of TGF-β in endothelial cells [95]. In what looks like classic bystander signaling, inflammatory responses are attenuated at localized doses <1 Gy, while anti-infection immunity is simultaneously made more effective by these doses [88,95].

### 2.6. Radiation Hormesis: When Small Amounts Bring a Benefit

The final phenomenon challenging the mathematics of classical radiation biology is radiation hormesis. “Hormesis”, as a term, signifies stimulating or beneficial effects obtained by administering a small amount of poison to a living being [96]. Already, before World War I, French scientists had observed stimulating effects on the growth of plant seedlings under very low X-ray exposure [97]; these observations were later verified in Japan with rice seedlings [95,98]. In the period preceding World War II and the atomic bombings of Hiroshima and Nagasaki, strong experimental evidence accumulated that protracted low-dose exposures, typically in the 50–200 mGy range, and never exceeding 500 mGy, were beneficial for plant growth and fungal growth [96,97,98,99,100]. Similarly, insects, particularly two species of flour beetles (*Tribolium confusum* Jacquelin du Val, 1863 and *Tribolium castaneum* Herbst, 1797) were observed to have extended survival when irradiated by up to 0.6 Gy, or 1/75th of a lethal dose, of 50 kVp X-rays [101,102]. Similarly, low-to-medium doses of X-rays, typically not exceeding 1 Gy, were found to be beneficial in the treatment of inflammatory and degenerative diseases in human patients, as discussed in the previous section [88,89,90,91,92,103]. Nevertheless, in the period after World War II, characterized by the observed horror of the atomic bombings in 1945 and the pervasive fear of nuclear warfare, studies of radiation hormesis became marginalized and the possibility of beneficial low-dose effects of radiation has never been included in calculations of risk for the purposes of radioprotection [99,100,103].

## 3. Conclusions: Are We Asking the Right Questions?

The previous two sections discussed (I) the dependence of classical radiation biology on the mathematical realities of DNA damage and repair and (II) the main challenges to classical radiobiology and radioprotection, namely (1) inter-specific and inter-individual variations in radiosensitivity, even in closely related organisms; (2) “distal” effects of radiation exposure; (3) non-targeted effects of radiation, including bystander effects (RIBE) and trans-generational genomic instability; (4) the genetics of normal-tissue radiosensitivity and susceptibility to cancer induction, only as the visible tip of the iceberg of “individual radiosensitivity”; and (5) immune system modulation by radiation.

Here, it is obligatory to state, again, that modern radiation biology accounts for the existence of non-targeted effects of radiation such as RIBE and genomic instability, as well as radiation hormesis; nevertheless, all current risk estimates are derived from the LNT model, which is based on the mathematics of classical radiation biology.

But how is all of this relevant?

It is conceivable that we, as scientists and radiation biologists, may be overly entrenched in what we know because we know a lot—the discipline has been advancing in leaps and bounds over the five last decades. Ionizing radiation is probably the best studied genotoxin, and the “radiobiology scene” is, likewise, among the life science communities that seem most proficient in their subject matter. Sometimes, though, we approach open questions with dogmatic answers. To some extent, we have lost the ability to be naïve. For example:We know that acute radiation responses affect stem cells—in the bone marrow, in the crypts of the small intestine, and in the skin. At the same time, skeletal muscle and connective tissue (with exactly the same genome) appear almost unaffected. We take this for granted, but is it unavoidable? Dr. Theodor Fliedner, one of the pioneers of radiation hematology, called ARS “a disturbance in cellular kinetics”. We treat ARS conservatively with growth factors, BM transplant, and supportive care. Is it not possible to target ARS by rescuing stem cells via strategies as diverse as “stemness inhibitors” or “signaling inhibitors”? Keeping in mind the diverse and significant roles of inflammation signaling, “bystander signaling”, DNA damage checkpoints, pro/antiapoptotic receptors, and cell differentiation, it seems fully conceivable that one day we will be able to manage ARS better with a cocktail of small molecules administered in the first 1–2 days post-irradiation.The last two decades have showcased the progress of immuno-oncology. First-generation immune checkpoint inhibitors (e.g., monoclonal antibodies (mABs) against PD-1, PD-L1, CTLA-4) are revolutionizing the clinician’s approach, though only 20–40% of patients benefit [104,105], while CAR-T therapies offer hope for refractory blood cancers; oncolytic viruses and anticancer vaccines are in corporate pipelines. Keeping in mind the immunomodulatory properties of IR, can we develop radio-immuno-oncology [106]? Similarly to the above idea for ARS, it is conceivable that small-molecule targeting of inflammation signaling, “bystander signaling”, DNA damage checkpoints, and pro/antiapoptotic cascades will lead to “precision radiation oncology” with better safety margins and more extensive cancer targeting.Despite what we think we know about radiosensitivity and radioresistance, we still cannot provide a coherent answer to the question of what makes an organism more or less susceptible to IR damage. Is it time to look back at physiology, signaling, and functional genomics in a wider array of species? Or, perhaps, to employ contemporary tools like machine learning, Markov models, and omics approaches to assess the molecular mechanics of individual organisms and cell types and to solve the secrets of radiation biology and of cell and molecular biology in general?

In the present climate of at-risk academic funding and precarity of smaller scientific fields (with each smaller discipline being declared “moribund” at least once in a decade), it is still worthy to remember that we have not yet “solved” radiation biology and that there is still a long way ahead of us.

## Figures and Tables

**Table 1 ijms-26-00257-t001:** Selected heritable syndromes inducing normal-tissue (NT) radiosensitivity; based on [1] and [80,81,82,83], gene ID numbers obtained from the NCBI GENE database.

Human Disease	Affected Gene(s)	Affected System(s)	Etiology and Clinical Manifestation
**Artemis deficiency**	*DCLRE1C* **(NCBI GENE ID 64421)**	NHEJ, V(D)J recombination	Rare autosomal recessive defect in Artemis. Radiosensitive SCID which results in a T-B-NK+ phenotype [80,82]
**Ataxia-telangiectasia (AT)**	*ATM* **(NCBI GENE ID 472)**	DNA damage response, NHEJ, HR	Autosomal recessive defect in ATM. Neurodegeneration resulting in ataxia, immune deficiency (IgA, IgM, and IgG subclasses), cancer predispositions (leukemia, lymphoma), telangiectasias [1,80]
**Ataxia-telangiectasia-like disorder (ATLD)**	*MRE11A* **(NCBI GENE ID 4361)**	HR, Alt-NHEJ	Very rare recessive defect or inactivation of ATM or MRE11. Neurodegenerative effects similar to AT but without immune deficiency and telangiectasias [82,83]
**Bloom syndrome ***	*BLM* **(NCBI GENE ID 641)**	HR, damage sensing, translesion DNA synthesis	Autosomal recessive defect of BLM. Short stature, erythematous skin rash upon sun exposure, genomic instability, premature aging, predisposition to cancer [80,81,82,83]
**Cernunnos/XLF deficiency syndrome**	*NHEJ1* **(NCBI GENE ID 79840)**	NHEJ, V(D)J recombination	Recessive defect of NHEJ1. Combined immunodeficiency, microcephaly, growth retardation, T- and B-cell lymphopenia [82,83]
**Cockayne syndrome**	*ERCC6* **(NCBI GENE ID 2074)***/ERCC8* **(NCBI GENE ID 1161)**	Nucleotide excision repair (NER)	Recessive defect of ERCC6 or ERCC8. Neurodegeneration, progeria, photosensitivity but no excess cancer risk [1,82,83]
**Fanconi’s anemia (FA)**	An amount of 21 genes, including *FANCA* **(NCBI GENE ID 2175)***, FANCB* **(NCBI GENE ID 2187)***, FANCC* **(NCBI GENE ID 2176)***, BRCA1/2* **(NCBI GENE ID 1161/675)***, RAD51C* **(NCBI GENE ID 5889)**	DNA damage sensing, nucleotide excision repair (NER)	Autosomal or X-linked recessive defect in nucleotide excision repair. Bone marrow failure, increased risk of hematological malignancies (AML and myelodysplastic syndrome, MDS) and liver tumors [1,82,83]
**Hutchinson-Gilford Progeria (HGPS)**	Lamin A/C **(NCBI GENE ID 4000)**	Nuclear architecture, HR and, NHEJ	Autosomal dominant, usually spontaneous mutation in the LMNA gene. Premature aging phenotype, cardiovascular damage, atherosclerosis, alopecia, genomic instability [80,83,84]
**Ligase IV syndrome**	*LIG4* **(NCBI GENE ID 3981)**	NHEJ, V(D)J recombination	Rare recessive mutation leading to functional reduction of Ligase IV activity. B- and T-cell lymphopenia, sometimes SCID, dysmorphic features, high incidence of cancers [82,83]
**Nijmegen breakage syndrome**	*NBN* **(NCBI GENE ID 7652)**	HR	Autosomal recessive inheritance. Mutation in NBN leading to progressive microcephaly, immunodeficiency, early growth retardation, very high incidence of cancers (mainly leukemia/lymphoma) [1,80,82]
**Rothmund–Thomson syndrome ***	*RECQL4* **(NCBI GENE ID 9401)**	HR, damage sensing, translesion DNA synthesis	Autosomal recessive defect in the RECQL4 DNA helicase. Sun-sensitive erythema, juvenile cataracts, premature aging, susceptibility to cancer (particularly osteosarcomas) [82,83]
**Seckel syndrome**	*ATR* **(NCBI GENE ID 545)**/*CENPJ* **(NCBI GENE ID 55835)**/unknown	HR, repair resolution, G2/M checkpoint arrest	Rare, autosomal recessive loss of function of ATR or CENPJ, sometimes unknown causation. Low birth weight, dwarfism, microcephaly with intellectual disability [80,82,83]
**Werner syndrome ***	*WRN* **(NCBI GENE ID 7486)**	NHEJ, resolution of HR repair, BER	Autosomal recessive deficiency of WRN expression. Rapid aging after puberty, vascular damage, skin conditions, susceptibility to malignant melanoma and rare cancers (hemangioma, meningiomas, MDS) [80,81,82,83]
**Xeroderma pigmentosum spectrum**	*XPA* **(NCBI GENE ID 7507)***, XPB* **(NCBI GENE ID 2071)***, XPC* **(NCBI GENE ID 7508)***, ERCC4/5/6* **(NCBI GENE IDs 2072, 2073, 2074)***, DDB2* **(NCBI GENE ID 1643)***, POLH* **(NCBI GENE ID 5429)**	NER, DNA damage recognition	Familial, autosomal recessive loss of function on one of the XP-associated genes with varying severity of disease presentation. Skin lesions; high risk of skin cancer, with about half of patients having skin cancer by age 10 without preventative efforts; and cataracts [1,80,81,82,83]

* Bloom syndrome, Werner syndrome, and Rothmund–Thomson syndrome, although separate conditions with distinct clinical manifestation are based on genetic homologs of the bacterial RecQ helicase.

**Table 2 ijms-26-00257-t002:** Selected heritable syndromes inducing increased susceptibility to cancer; gene ID numbers obtained from the NCBI GENE database.

Human Disease	Affected Gene(s)	Affected System(s)	Etiology and Clinical Manifestation
**Adenomatous polyposis coli (APC)**	*APC* **(NCBI GENE ID 324)**	Cell division checkpoint	Familial autosomal dominant deletion in one copy of the APC gene. Benign colonic adenomas progressing to cancer (7% risk by age 21, rising to 87% by age 45 and 93% by age 50) [1,80,82]
**Cowden syndrome**	*PTEN* **(NCBI GENE ID 5728)**	Metabolism/cell proliferation checkpoint	Autosomal dominant mutation in PTEN, leading to dysregulation of the mTOR pathway. Hamartomas, excess risk of breast, thyroid, uterine, skin, and other cancers [80,83]
**Gorlin syndrome (nevoid basal-cell carcinoma syndrome)**	*PTCH1* **(NCBI GENE ID 5727)**	Cell division and embryogenesis checkpoint	Autosomal dominant mutation of the *PTCH* (Patched) gene. Unusual facial features, predisposition for skin and bone cancers, benign osteogenic cysts [1,80,83]
**Hereditary breast cancer**	*BRCA1/BRCA2* **(NCBI GENE IDs 672/675)**	DNA damage checkpoints, HR	Dominant defective single copy of *BRCA1* or *BRCA2* resulting in probable loss of the other gene copy. High risk of hereditary breast cancer, with up to 70% risk of developing cancer by age 80 [1,80,82]
**Hereditary retinoblastoma**	*RB1* **(NCBI GENE ID 5925)**	Cell cycle inhibition, DNA methylation	Autosomal dominant defective *RB1* tumor suppressor gene. Very high risk for retinoblastoma and bone cancer; one defective copy induces high-risk phenotype [1,80,82]
**Li–Fraumeni syndrome**	*TP53* **(NCBI GENE ID 7157)**	DNA damage checkpoint, HR	Autosomal dominant defect in the *TP53* tumor suppressor gene. Early-onset sarcomas, breast cancer, central nervous system (CNS) tumors, leukemias, and adrenocortical carcinomas [80,82,83]
**Lynch syndrome (hereditary nonpolyposis colorectal cancer (HNPCC))**	*MLH 1,3,* **(NCBI GENE IDs 4292/27030)** and *MSH 2,6* **(NCBI GENE IDs 4436/2956)**	Mismatch repair (MMR)	Autosomal dominant defect in mismatch repair (MMR). High risk of colorectal, endometrial, hepatobiliary, urinary, and brain cancers [80,82,83]
**Von Hippel–Lindau disease**	*VHL* **(NCBI GENE ID 7428)**	Regulation of hypoxia-inducible transcriptome	Dominant, inactivating mutation in the *VHL* tumor suppressor gene. Locomotor problems, visceral cysts, and benign tumors with potential for subsequent malignant transformation; increased risk of kidney cancer [82,83].

## Data Availability

The original contributions presented in the study are included in the article; further inquiries can be directed to the corresponding author.

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
