# Peer review of "The Significance of the Response: Beyond the Mechanics of DNA Damage and Repair—Physiological, Genetic, and Systemic Aspects of Radiosensitivity in Higher Organisms"

_ijms, 2024, doi:10.3390/ijms26010257_

Round 1

Reviewer 1 Report

Comments and Suggestions for Authors

In this manuscript entitled” The significance of the response: beyond the mechanics of DNA damage and repair – physiological, genetic, and systemic aspects of radiosensitivity in higher organisms, the author provides a comprehensive review of the classical radiation biology focused on DNA damage and repair. Additionally, the author highlights the challenges to the classical radiation biology, such as species-specific radio-sensitivity variations at individual, tissue/organ, and cellular levels, as well as abscopal effect and non-targeted effects. However, it is important to note that classical radiation biology has expanded its scope from solely focusing on DNA damage and repair to encompassing all aspects of DNA damage responses (DDR), including damage sensing, signaling, cell cycle control, repair, and apoptosis. In my opinion, these physiological, genetic, and systemic factors mentioned by author influence radio-sensitivity more possibly through their regulation of DDR processes. Therefore, rather than challenging classical radiation biology principles they actually enrich our understanding in classical radiation biology. I hopes that the revised version should consider incorporating this this viewpoint.

Generally speaking, the manuscript is difficult to read due to issues with expression and logicality. Additionally, certain content, such as those concerning mtDNA, is not relevant to the topic at hand.  

Comments on the Quality of English Language

The language should be greatly improved. 

Author Response

First, I would like to thank the Reviewer for taking the time to assess the current work; I have addressed most if not all suggestions in the manuscript, and would like to respond to the following points:

Reviewer comment 1:
In this manuscript entitled” The significance of the response: beyond the mechanics of DNA damage and repair – physiological, genetic, and systemic aspects of radiosensitivity in higher organisms, the author provides a comprehensive review of the classical radiation biology focused on DNA damage and repair. Additionally, the author highlights the challenges to the classical radiation biology, such as species-specific radio-sensitivity variations at individual, tissue/organ, and cellular levels, as well as abscopal effect and non-targeted effects. However, it is important to note that classical radiation biology has expanded its scope from solely focusing on DNA damage and repair to encompassing all aspects of DNA damage responses (DDR), including damage sensing, signaling, cell cycle control, repair, and apoptosis. 
Author's Response 1: While I agree that "modern radiation biology" takes into account non-targeted effects such as Bystander effects (RIBE) and genomic instability (GI), as well as abscopal effects, cell signaling/apoptosis, etc., I have to mention that these phenomena were not originally a part of "classical radiation biology", which is precisely based on cell survival curves and the mathematics of DNA damage and repair. I have expanded upon DNA damage responses (DDR), which are also addressed indirectly throughout the manuscript. "Classical radiation biology" is also the basis for current radioprotection and risk estimates (via the LNT model), which do not take into account RIBE, GI, or hormesis. I have now included this information in both the introduction and the conclusion.

Reviewer comment 2: In my opinion, these physiological, genetic, and systemic factors mentioned by author influence radio-sensitivity more possibly through their regulation of DDR processes. Therefore, rather than challenging classical radiation biology principles they actually enrich our understanding in classical radiation biology. I hopes that the revised version should consider incorporating this this viewpoint.
Author's Response 2: This is addressed in the revised version. Rather than challenging either "classical" or "modern" radiation biology, the objective of the current article is to provide and substantiate new potential venues for further radiobiological research, for instance 1) detailed inquiries into comparative molecular physiology and cell signalling; 2) radio-immunooncology, or 3) using "machine learning", Markov models, and other AI applications to infer new data on key aspects of radiosensitivity at the system, organism, and population levels.

Reviewer comment 3: Generally speaking, the manuscript is difficult to read due to issues with expression and logicality. Additionally, certain content, such as those concerning mtDNA, is not relevant to the topic at hand.  
Author's Response 3: The manuscript is written in good academic English; I have reviewed expressions and logicality throughout the text, and provided clarifications where necessary. mtDNA information is included in the text for the purpose of comprehensive analysis, and because mitochondrial DNA damage and related signaling has been discussed previously by other authors in the context of radiation biology.

Reviewer 2 Report

Comments and Suggestions for Authors

Comments to author:

The manuscript of a review article, which was written by Dr. Peter Vladislavov Ostoich, is interesting, discussing the radiosensitivity of organisms from the aspect of biological mechanisms against DNA damage from various stresses. This article is also related to this year's Nobel Peace Prize, making it a very timely topic.

Recommendation: Minor revision

General comments

  This is a review article includes descriptions of details of the mammalian DNA repair systems. I thought that a timetable for the history of DNA repair studies would be better added. That will greatly help readers’ comprehension. Moreover, I encourage the author to describe more about differences of exposure to different types of irradiations, alpha, beta, and gamma.

Minor comments

P6, L293, P8, L324: Do not include references in the Title of the Table. They should be indicated for each description in the Table.

P6-9, Table 1 and 2: Condition: Human diseases?

P6-9, Table 1 and 2: Affected systems need to be explained or defined. For example, the differences between NHEJ and Alt-NHEJ should be described in the legend.

P6-9, Table 1 and 2: Clinical manifestation should be clearly described. For example, why ATM gene is attributed for the pathogenesis of the AT? Is it mutation, or suppression, or up-regulation of the ATM?

Table 1 and 2 could be combined.                       

P9: Rb and p53: RB1 and TP53

GENE ID (from the database of NCBI, GENE) should be indicated.

In this review article, no descriptions about low dose irradiation effects are included. High dose will be harmful but low dose rather cause beneficial effects. This concept, which is referred to as hormesis, is important when discussing radiation.

Author Response

First, I would like to thank the Reviewer for the detailed review, knowledgeable feedback and constructive suggestions! I have tried to include all of the suggested revisions in the updated manuscript.

Reviewer comment 1:
This is a review article includes descriptions of details of the mammalian DNA repair systems. I thought that a timetable for the history of DNA repair studies would be better added. That will greatly help readers’ comprehension. Moreover, I encourage the author to describe more about differences of exposure to different types of irradiations, alpha, beta, and gamma.
Author's Response 1: I have added more information on the mammalian DNA repair systems in the introduction, as well as making Table 1 and Table 2 more understandable. The history of DNA repair is a very interesting topic, but it has already been addressed elsewhere, for example:
1) Friedberg EC. A brief history of the DNA repair field. Nature: Cell research. 2008 Jan;18(1):3-7.
2) Saini N. The journey of DNA repair. Trends in cancer. 2015 Dec 1;1(4):215-6.
While it is tempting to include a timeline of DNA repair discoveries, this is a very big topic, and expands away from mammals into other organisms such as bacteria, yeast (particularly S. cerevisae), and plants (beans, peas, maize, and others). It is an excellent idea for a future paper, though!
I have added more material on the effects of exposure to different radiation types.

Reviewer comment 2: In this review article, no descriptions about low dose irradiation effects are included. High dose will be harmful but low dose rather cause beneficial effects. This concept, which is referred to as hormesis, is important when discussing radiation.
Author's Response 2: I agree, and I have included an entire new section on radiation hormesis.

Reviewer's minor comments:

P6, L293, P8, L324: Do not include references in the Title of the Table. They should be indicated for each description in the Table.
Author's action: corrected the table as suggested.

P6-9, Table 1 and 2: Condition: Human diseases?
Author's action: corrected the table as suggested.

P6-9, Table 1 and 2: Affected systems need to be explained or defined. For example, the differences between NHEJ and Alt-NHEJ should be described in the legend.
Author's action: corrected according to the suggestions - now DNA repair systems are expanded upon in the introduction.

P6-9, Table 1 and 2: Clinical manifestation should be clearly described. For example, why ATM gene is attributed for the pathogenesis of the AT? Is it mutation, or suppression, or up-regulation of the ATM?
Author's action: corrected the table, now the disease etiology (genetic cause) and clinical manifestations are revised and described in more detail.

Table 1 and 2 could be combined.  
Author's action: I have decided to leave them separate, because there are two key differences between the tables:
1) In Table 1, the diseases cause increased normal tissue sensitivity to radiation, while in Table 2 the diseases cause increased cancer risk without increasing the radiosensitivity of normal tissues.
2) In Table 1, the diseases are mostly recessively inherited, and the genes affected are mostly DNA DSB repair systems; in Table 2, the diseases are mostly autosomal, dominantly inherited, and affected systems are "non-DSB repair" and cell signalling/DDR.

P9: Rb and p53: RB1 and TP53
Author's action:
corrected as suggested.

GENE ID (from the database of NCBI, GENE) should be indicated.
Author's action: corrected as suggested.

Round 2

Reviewer 1 Report

Comments and Suggestions for Authors

I think the current status is acceptable for publication following the enhancement of language.

Comments on the Quality of English Language

The language should be refined to enhance clarity and readability.